# A distinct four-value blood signature of pyrexia under combination therapy of malignant melanoma with dabrafenib and trametinib evidenced by an algorithm-defined pyrexia score

Hannah Schaefer[1], Albert Rübben[1,2,3]*, André Esser[4], Arturo Araujo[5], Oana-Diana Persa[3,6¤], Marike Leijs[1,2,3]

1 Department of Dermatology, RWTH Aachen University Hospital, Aachen, Germany, 2 Department of Dermatology, St. Nikolaus Hospital, Eupen, Belgium, 3 Center for Integrated Oncology, CIO ABCD, Aachen, Bonn, Cologne, Düsseldorf, Germany, 4 Department of Occupational, Social and Environmental Medicine, RWTH Aachen University Hospital, Aachen, Germany, 5 Department of Media, Culture and Language, University of Roehampton, London, United Kingdom, 6 Department of Dermatology and Venereology, University Hospital Cologne, University of Cologne, Cologne, Germany

¤ Current address: Department of Dermatology and Allergy Biederstein, Technical University of Munich, Munich, Germany

* albert.ruebben@post.rwth-aachen.de

**Data Availability Statement:** We have deposited the processing of the pyrexia score as well as the anonymized datasets containing time series of

## Abstract

Pyrexia is a frequent adverse event of BRAF/MEK-inhibitor combination therapy in patients with metastasized malignant melanoma (MM). The study's objective was to identify laboratory changes which might correlate with the appearance of pyrexia. Initially, data of 38 MM patients treated with dabrafenib plus trametinib, of which 14 patients developed pyrexia, were analysed retrospectively. Graphical visualization of time series of laboratory values suggested that a rise in C-reactive-protein, in parallel with a fall of leukocytes and thrombocytes, were indicative of pyrexia. Additionally, statistical analysis showed a significant correlation between lactate dehydrogenase (LDH) and pyrexia. An algorithm based on these observations was designed using a deductive and heuristic approach in order to calculate a pyrexia score (PS) for each laboratory assessment in treated patients. A second independent data set of 28 MM patients, 8 with pyrexia, was used for the validation of the algorithm. PS based on the four parameters CRP, LDH, leukocyte and thrombocyte numbers, were statistically significantly higher in pyrexia patients, differentiated between groups (F = 20.8; p = <0.0001) and showed a significant predictive value for the diagnosis of pyrexia (F = 6.24; p = 0.013). We provide first evidence that pyrexia in patients treated with BRAF/MEK-blockade can be identified by an algorithm that calculates a score.

## Introduction

Within the last decade, cutaneous melanoma incidence has been rising steadily worldwide [1, 2]. 40–60% of all patients with malignant melanoma show an activating mutation of the

leukocyte and thrombocyte counts as well as of CRP and LDH serum levels of all patients in the open access datadryad.org data repository as an excel file (https://doi.org/10.5061/dryad.xpnvx0kjj).

**Funding:** The author received no specific funding for this work.

**Competing interests:** The authors have declared that no competing interests exist.

serine-threonine kinase B-RAF (BRAF) [3, 4], the most common BRAF mutation being the V600E point mutation (T→ A nucleotide change). Activating mutations at BRAF V600 lead to a constitutive activation of the MAP kinase signaling pathway (RAS-RAF-MEK-ERK), making these proteins an attractive target for therapies [3].

Patients treated with the combination of the selective BRAF inhibitor dabrafenib and the MEK inhibitor trametinib show a longer progression-free survival (11 months vs 8,8 months), and an improvement of the overall survival (25,1 months vs 18,7 months), as compared to a monotherapy with a BRAF inhibitor [2, 5, 6]. In the US and in Europe, combination therapy with a BRAF-inhibitor and a MEK-inhibitor is approved for patients with unresectable BRAF-V600-mutated melanoma as well as for adjuvant treatment in stage III melanoma [2, 5–7]. Other combinations of approved BRAF/MEK inhibitors such as vemurafenib and cobimetinib or encorafenib and binimetinib have demonstrated similar efficacies [4].

Mostly similar negative side effects can be observed in mono- and combination therapies with BRAF/MEK inhibitors, but pyrexia seems to be particularly frequent when using the combination of dabrafenib and trametinib. Considering all grades, it has been reported in up to 59% of patients treated with dabrafenib and trametinib [8, 9]. Despite these pioneering efforts to develop a therapy for melanoma patients, pyrexia is one of the most common reasons, not only for dose reduction and interruption, but also for a complete therapy discontinuation [7, 10]. Furthermore, under BRAF/MEK inhibition, myelosuppression has been observed [11–13]. Some patients may also experience more than one pyrexia episode and the episodes might be accompanied by hypotension or complicated by neutropenic sepsis [14].

The mechanisms inducing pyrexia under BRAF/MEK-inhibition are not fully understood yet and no clinical marker seems to be available which reliably predicts which patients will develop pyrexia [15]. Still, in previous publications it could be demonstrated that pyrexia under BRAF/MEK inhibition is associated with an increase of acute phase proteins such as CRP and procalcitonin, with an increase of proinflammatory cytokines such as interleukin-1beta (IL-1beta) and interleukin-6, a decrease of leukocytes and granulocytes and with aberrations of the coagulation system [11–13, 16, 17].

In several patients treated at the Department of Dermatology at the University Hospital of the RWTH Aachen, Germany, we observed an increase of C-reactive protein (CRP) in parallel with a decrease of leukocyte and thrombocyte counts at the time of pyrexia and sometimes already before pyrexia (Fig 1). Starting from this observation which suggested that the

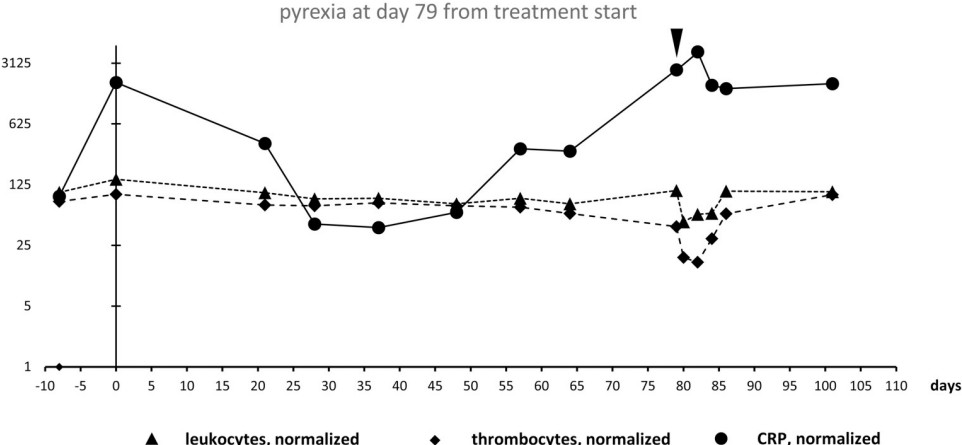

**Fig 1. Time series of a patient with pyrexia at day 79 from treatment start, semi-logarithmic scale.** CRP was normalized to 100 = 2.5 mg/l (value CRP patient/2.5 * 100), leukocyte count was normalized to 100 = 6.5 /nl (for male) & 6 /nl (for female), thrombocyte count was normalized to 100 = 275 /nl. Triangle indicates onset of pyrexia.

synchrony of observed laboratory value changes might be predictive as well, we initiated a retrospective study to identify pattern changes within routine laboratory values which could diagnose as well as predict the onset of pyrexia in order to improve management of melanoma patients. Besides traditional statistics used for detecting associations between laboratory values before pyrexia and at the time around pyrexia, an algorithm-based approach was developed for the detection of pyrexia and its results were compared to the statistical analysis of individual laboratory values.

## Materials and methods

### Ethics statement

This retrospective study was conducted in accordance with the guidelines of the Declaration of Helsinki. Ethical review and approval were obtained and confirmed in advance by the local ethics committee of the University Hospital of the RWTH Aachen (ethic vote 257/19). The ethics committee did not require informed consent of the patients as the study was strictly retrospective and only analysed available routine laboratory data of the patients without assessing the efficacy of treatments and without contacting of patients or relatives. Prior to statistical data analysis, all anamnestic information was anonymized and time date indications were expressed without a specific calendar date.

### Study population

We conducted a retrospective analysis of medical records of 38 patients (27–84 years) including 24 (65%) non-pyrexic (17 male & 7 female) and 14 pyrexic patients (8 male & 6 female), who received combination therapy with dabrafenib plus trametinib at the Department of Dermatology of the University Hospital of the RWTH Aachen from 02/2015 until 04/2020.

Therapy consisted of oral dabrafenib (150 mg twice daily (BID)) and oral trametinib (2mg once a day (QD)). In some patients, the initial dabrafenib dose was reduced to 150 mg per day during the first three days of treatment followed by an evaluation of serum amylase and/or lipase in order to avoid severe pancreatitis which can be associated with this treatment. Start of trametinib treatment was delayed by up to one week in some patients when cardiologic evaluations were not available but inhibition therapy with dabrafenib could not be postponed. In all patients, pyrexia developed under combination therapy. Data from this study population was used for initial statistical analysis and the development of the pyrexia score algorithm.

A second independent data set was generated from a retrospective analysis of 28 patients including 8 additional patients with pyrexia and 20 additional patients without pyrexia from the Department of Dermatology of the University Hospital of the RWTH Aachen (n = 5) and from the Department of Dermatology and Venereology, Cologne University Hospital, Germany (n = 23) who were treated with dabrafenib and trametinib from 09/2015-05/2022. This second data set was used to validate the pyrexia score algorithm developed by the above-described retrospective analysis and was, therefore, blinded and not adjusted for age or sex and consisted only of laboratory data.

The patient's case histories and laboratory values were obtained from the institution's digital patient information system of the University Hospital of the RWTH Aachen and from the Department of Dermatology and Venereology, Cologne University Hospital, Germany. Data acquisition by the treating physician and statistical data analysis were strictly separated in order to ensure patients' data protection.

Inclusion criteria for all patients were age (>18 years) and a confirmed diagnosis of metastatic cutaneous melanoma (stage III or stage IV) treated with dabrafenib and trametinib. Pyrexia was defined as an oral temperature of 38.5˚C ($\geq$101.3˚F) or higher in the absence of

any clinical or microbiological evidence of infection. A pyrexic event was deemed to have resolved after a 24h period of temperature of 37.5˚C or less. Patients who had missing data in our institution's records regarding their treatment regimen and/or side effects were excluded from the statistical analysis.

## Laboratory parameters

Biochemical and haematological parameters obtained from the routine laboratory evaluations of all 38 patients were examined for the initial statistical analysis and for the development of the pyrexia score algorithm. Parameters included differential blood count (leukocytes, neutrophils, basophils and eosinophils, lymphocytes, monocytes), erythrocytes, haemoglobin, haematocrit, mean corpuscular volume (MCV), mean corpuscular haemoglobin (MCH), mean corpuscular haemoglobin concentration (MCHC), thrombocytes, glucose, lipase, aspartate transaminase (AST), alanine aminotransferase (ALT), gamma-glutamyl transferase (gamma-GT), lactate dehydrogenase (LDH), creatine kinase (CK), uric acid, urea, urea-creatinine quotient, Creatinine, glomerular filtration rate (GFR), C-reactive protein (CRP), protein S100, interleukin 6 (IL6) and thyroid stimulating hormone (TSH). For the validation data set, only laboratory values for leukocyte and thrombocyte counts, LDH and CRP during BRAF/MEK-inhibitor treatment were obtained from 28 patients. For normalized data (percentages), the original laboratory data was divided by the sex-specific mean value of the respective parameter and the result was multiplied by 100.

## Database

Using Microsoft Excel (MS Excel 2016, Version 15.11), anonymized data sets were created. All data relating to the respective patient history were compiled in an anamnestic table. The extracted variables included age, sex, tumor stage (at time of initial diagnosis and initiation of therapy), tumor classification (according to the TNM system), time of initial diagnosis and initiation of therapy, location of the primary tumor, metastases, secondary diagnoses, medication duration and side effects of therapy. Oncology progress notes were reviewed to identify any inflammatory side effects, time to their onset, and associated management.

The patient's laboratory data were recorded in a different database and assigned to a time axis, starting from the first day of therapy initiation (T0 = initiation of therapy). Each pyrexic event was retrospectively analysed and assigned to the start of treatment (or to the restart of treatment if it was discontinued). The total number of pyrexic events was recorded and each event was analysed separately. The result of the occurrence was binary dichotomized with regard to its expression whether there were inflammatory side effects (pyrexia) or they were absent (no pyrexia). In patients with recurrence of a certain side effect, time to the first presentation was used as time to side effect onset.

## Time cluster analysis

Since it was a retrospective study, laboratory data were determined irregularly and with varying frequency for each patient. To avoid the problem of heterogeneous observation units and to create a better comparability within the cohort, a time cluster consisting of four individual data points was created. The time clusters were then treated as a time dependent covariate in the statistical analyses of the laboratory data. Every cluster included exactly one data point for every patient, in order to have four different data values at comparable time points to include in our statistics.

Laboratory values of patients who did not develop pyrexia were also assigned to the four time clusters based on the determination of the mean time value of the occurrence of pyrexia

in our data set, thereby assuming a time at which a pyrexia might have occurred in these patients. This strategy was used in order to rule out that the suspected laboratory values might change by a similar mode in time regardless of pyrexia.

The four time clusters covered data points within the time phases, i.e., last laboratory evaluation before the start of therapy (1st cluster: pre-T), first data after the start of therapy (2nd cluster: post-T), last laboratory evaluation before the onset of pyrexia (3rd cluster: pre- P) and first laboratory values after the onset of pyrexia but before any treatment of pyrexia (4th cluster: post-P). Cluster 1 covered 33 days (from T-1 to T-33), cluster 2 covered 21 days (from T1-T21), cluster 3 covered 23 days (from P-1 to P-23) and cluster 4 covered 28 days from (P0 to P28). Laboratory values for cluster 1 were missing in 4 patients, laboratory values for cluster 2 were missing in 4 patients, laboratory values for cluster 3 missing in 3 patients and laboratory values for cluster 4 in 2 patients.

## Statistical analysis

All analyses were conducted with the program SAS (SAS 9.4, SAS Institute, Cary, N.C., USA). The distribution of the independent parameters was classified through histogram comparison. Due to the results, as a first step a non-parametric correlation analysis (Spearman) between the development of pyrexia and laboratory data was conducted.

Descriptive statistics included percentage, frequency, median, and standard deviation were evaluated. General characteristics and tests of between-group homogeneity of the variables were analysed using the chi-square test, Wilcoxon test, ANOVA and t-test. The limit of significance was set for $\alpha$ at p = 0.05.

The impact of the individual blood parameters on the development of pyrexia was tested using a nested generalized mixed effects model. Due to the fact that the blood parameters as continuous variables revealed no saturated models, we utilized the quartiles of the particular blood parameter as a predictor. The time cluster was set as a random factor and the individual patient's ID was set as a nesting variable in dependence of the time cluster. Sex, age and the time cluster were included as covariates in the model, and we applied a post hoc Tukey test to adjust for repeated measurements. Additionally, we ran all models once more, including a term for the interaction of the blood parameter and the time cluster to detect if the effect of the blood parameter was time dependent.

The predictive value of the pyrexia score was also tested by a nested generalized linear mixed effects model. The individual patient's ID was set as a nesting variable, and the time span from onset of therapy until the blood sample was obtained was set as a random factor. We established a binomial regression with the pyrexia score as predictor, the time span variable as covariate and the development of the pyrexia as a dichotomous outcome. We applied this model to the respective scores.

To determine score thresholds for the development of a pyrexia, we ran a logistic regression model. We built a dummy variable and set this variable to zero for all participants without developing a pyrexia. The pyrexia score value of the other participants, who developed a pyrexia, was divided into quartiles and the particular number of the quartiles was applied for the dummy variable. The logistic regression model with developing pyrexia y/n as outcome and the dummy variable as predictor revealed the odds ratio and their confidence intervals for developing pyrexia in the particular quartiles of the score in comparison to the participants without developing pyrexia. The lower bounds were then adopted as thresholds, if the odds ratio was significantly increased. The same procedure was applied to the external validation cohort after the respective pyrexia scores were calculated.

## Development of a pyrexia score algorithm

Identification of previously unknown associations in medicine may be initiated by a single observation suggesting that a patient's characteristics, symptoms, clinical and laboratory data, side effects or treatment outcome in an individual patient or in a few treated patients could be correlated in a hitherto unreported fashion.

The starting point of the presented retrospective study was the observation in single patients of an increase in C-reactive protein (CRP) in parallel with a decrease of leukocyte and thrombocyte counts at the time of pyrexia and sometimes already before pyrexia (Fig 1). Graphical time-series visualization also demonstrated that available clinical routine data were sampled at uneven intervals ranging from single days to several weeks. In general, blood was drawn more often at the beginning of the treatment in order to exclude side effects such as pancreatitis and at the time of pyrexia as a direct consequence of the need to differentiate pyrexia from infection.

An algorithm is a process or set of rules that need to be followed in calculations or other problem-solving operations. In medicine, algorithms are widely used to define calculations of scores that may then be used for diagnosis, for objective evaluation of disease severity or for risk prediction. While some scores may be directly derived from statistical data, many scores in medicine have been developed following heuristic techniques.

We have adopted the idea of calculating a score by an algorithm for detection and prediction of pyrexia based on the following heuristic assumptions [18]:

- The data describe clinical time-series, but uneven temporal spacing of blood sampling in patients as well as missing values will probably limit the use of traditional mathematical methods for analysis of time series data [19].

- The visualization of laboratory data suggested that parallel changes of laboratory values during pyrexia were more important than single absolute values.

- From a clinician's point of view, laboratory values that demonstrate a trend like a continuous rise or fall over subsequent samplings, are considered more reliable than changes observed at a single time point.

- An algorithm may be optimized by an iterative and incremental optimization process, which by itself constitutes an algorithmic approach.

- An algorithm can integrate different values as well as temporal changes of values into one single decision or value. A single value, such as a score, may be subsequently analysed independently by traditional statistical methods.

Fig 2 describes the general structure of the algorithm developed for the calculation of the pyrexia score. Based on the results obtained by our statistical analysis (see results section), lactate dehydrogenase (LDH) was added to the other three laboratory values. Value 1 (V1) represents leukocyte count, V2 = thrombocyte count 2), V3 = LDH and V4 represents C-reactive protein (CRP).

The key heuristic assumption, that changes of laboratory values are probably more important than absolute values, was integrated into the algorithm by calculating the pyrexia score at the time of laboratory assessment (= tx) by comparing the individual values (V1, V2, V3, V4) at tx with the value of the previous blood sampling date at tx-1. The initial algorithm also contained threshold variables for a minimum change (F1-F4). For each value (V1-4), the first calculation (S1-S4) equals either 1 or 0. This basic algorithm was refined through a five-level iterative development following established Agile methodologies [20]. The heuristic

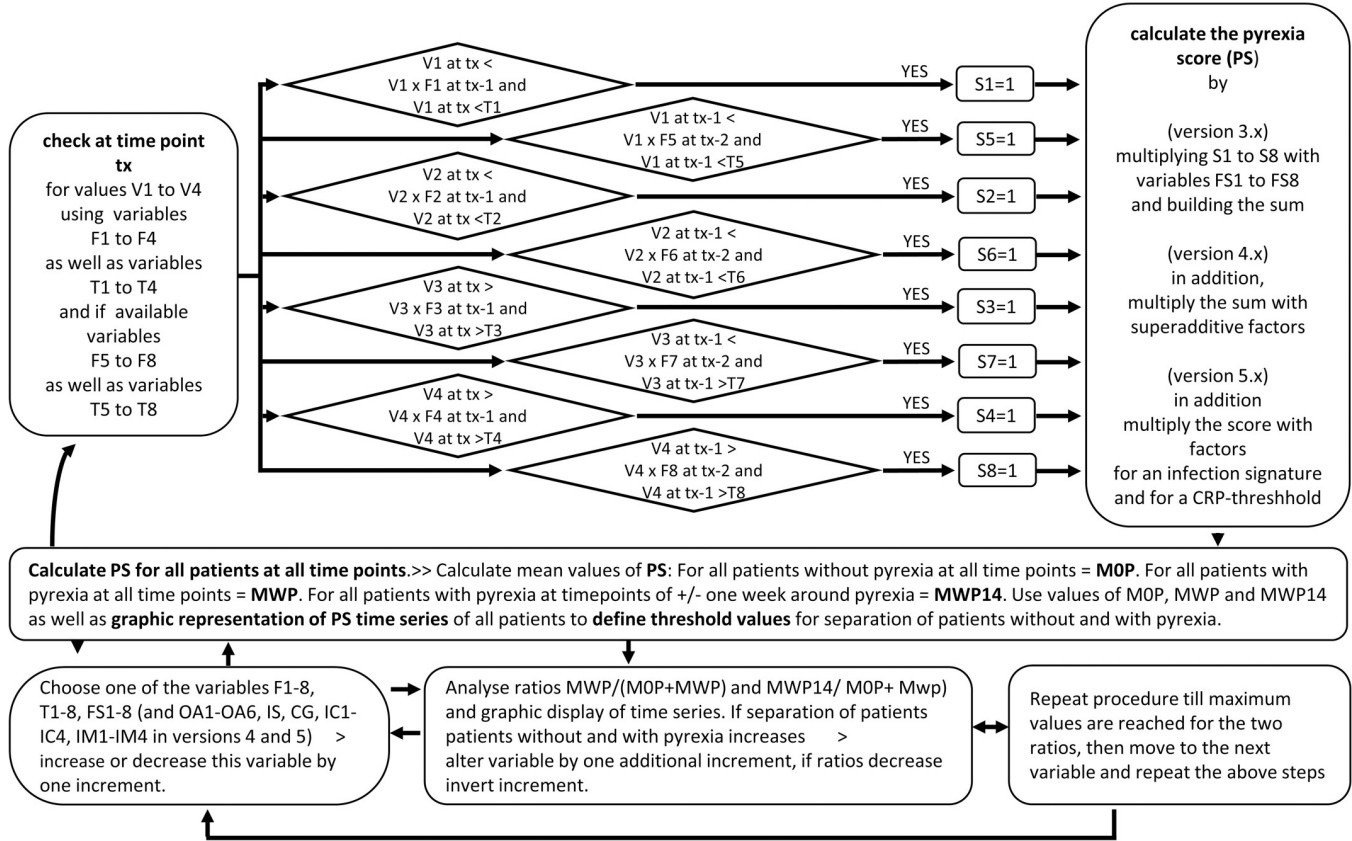

**Fig 2. Design and optimization of the pyrexia score algorithm.**

assumption, that a constant trend over subsequent blood samplings is a stronger indication than values determined at a single time point, was integrated into the algorithm by repeating the above calculations in version 2 of the algorithm for corresponding values V1-V4 at tx-1 and tx-2. In versions 1–2 of the algorithm, the assumption that pyrexia is characterized by a parallel change of these four values is reflected by the simplest possible calculation which consists of building a sum out of the eight calculations regarding V1 to V4 at tx as well as at tx-1, thus following Occam's Razor principle. In version 3, further threshold values for minimal (T1, T2) or maximum (T3, T4) values were introduced. Moreover, as it was unknown a priori which laboratory value would prove most important, initial calculations S1-S8 derived from values V1-V4 at tx as well as at tx-1 were multiplied by variables FS1-FS8 in version 3.

In version 4 of the algorithm, in order to explore whether combination of individual laboratory changes might be more indicative of pyrexia, further variables OA1-OA6 were introduced. At tx, in case of simultaneous decrease of leukocytes and thrombocytes (OA1), simultaneous decrease of leukocytes and increase of LDH (OA2), simultaneous decrease of leukocytes and increase of CRP (OA3), simultaneous decrease of thrombocytes and increase of LDH (OA4), simultaneous decrease of thrombocytes and increase of CRP (OA5) and simultaneous increase of CRP and LDH (OA6), the respective variables were attributed a value and the sum obtained out of the eight calculations regarding V1 to V4 at tx as well as at tx-1 was multiplied by OA1-OA6.

Version 5 of the algorithm introduced two additional calculations, one for a bacterial infection signature and one for a CRP-threshold. It was assumed that bacterial infection is

characterized by a simultaneous increase of CRP and an increase of leukocytes. Albeit with reduced sensitivity, an increase of LDH and thrombocytes has also been observed during bacterial infection [21]. For each leukocyte, CRP, thrombocyte and LDH value at t0, threshold variables were added for a minimum change IC1-IC4, as well as further threshold variables (IM1-IM4) were introduced for minimal values of leukocytes, CRP, thrombocytes and LDH. In case of a positive infection signature, PS were multiplied with variable IS ($<<1$), thus reducing the pyrexia score. It is further assumed that pyrexia is always associated with a CRP increase. If CRP at t0 does not exceed CRP value CT, PS is multiplied with variable GC ($<<1$), thus reducing pyrexia score. If CRP at t0 was equal or higher than CT or if no CRP value was recorded, this calculation reported 1. Supplementary file calculation-matrix-PS-5.0 contains an excel spreadsheet for calculation of PS-scores that also allows modification of variable values (see DYRAD repository: https://doi.org/10.5061/dryad.xpnvx0kjj).

In order to define the values of all the variables that would allow the strongest differentiation between patients not developing pyrexia and pyrexia patients (which would therefore also be able to best determine the onset of pyrexia), a heuristic, iterative and incremental optimization approach was chosen in versions 1.0–5.0 (Fig 2). This approach consisted in calculating pyrexia scores (PS) for all patients at all time points. Mean values of PS were then determined for all patients without pyrexia at all time points = M0P, for all patients with pyrexia at all time points = MWP as well as for all patients with pyrexia at time points of +/- one week around pyrexia = MWP14. These values were used to calculate the ratios MWP/(M0P+MWP) and MWP14/(M0P+ MWP). Besides, graphic displays of time series of patients without and with pyrexia were used to visualize the discriminative potential of the PS-algorithm. In an iterative process, one of the variables F1-8, T1-8, FS1-8, OA1-OA6, IS, CG, IC1-IC4, IM1-IM4 was increased or decreased by one increment and the above calculations were repeated. If ratios MWP/M0P and MWP14/M0P demonstrated an increase or graphic displays of time series demonstrated a better separation of patients without and with pyrexia, the same variable was altered by one additional increment and ratios and graphic displays were again determined. If the ratios decreased, the increment was reversed, and the procedure was repeated. The value of this variable was retained, and the next variable was chosen for the same procedure. Only one round of optimization was performed. Calculations and graphic display were conducted within an excel spreadsheet. Patients' data used for calculating the pyrexia scores as well an EXCEL-sheet that calculates the pyrexia score and which allows modification of the algorithm have been deposited as open access at the DYRAD repository under https://doi.org/10.5061/dryad.xpnvx0kjj.

## Results

### Descriptive statistics, first data set

We conducted a retrospective analysis of medical records of 38 patients (27–84 years) including 24 (65%) non-pyrexic (17 male & 7 female) and 14 pyrexic patients (8 male & 6 female), who received combination therapy with dabrafenib plus trametinib at the Department of Dermatology of the University Hospital of the RWTH Aachen, from 02/2015 until 04/2020. Patient sex was not correlated with development of pyrexia (chi-square test: p = 0.39).

During the observation period, 17 pyrexic events occurred in total. One patient developed two independent pyrexia episodes at intervals of 2 years, both of which were included in the statistics, because they had no causal connection and were regarded as independent events. One patient developed 3 pyrexic events, but episodes correlated with each other and were not independent, therefore only the first episode was included in the statistics. 86% (12 patients) of all pyrexic patients had only one event, 7% (1 patient) had 2 events and 7% (1 patient) had 3

events of pyrexia. The mean and median time from the beginning of therapy to onset of the first pyrexic event were 31 days and 26.5 days (range 5 days—79 days) and the mean and median time of duration of the pyrexic event were 2 days (range 1 day—4 days). Pyrexic events led to either temporary dose reduction or complete therapy interruption. When pyrexia had subsided, dabrafenib/trametinib therapy was either re-introduced or, in severe cases of recurrent pyrexia, therapy was finally discontinued. 8 out of 13 pyrexia patients (57%) with severe symptoms were treated with glucocorticoids.

The median age of all patients was 63 years. Patients who developed pyrexia were slightly older (median: 64 years) than those without any pyrexic event (median: 62 years). 28 patients (74%) had a stage IV MM, when starting therapy, 7 patients (18%) stage IIIC MM, 2 patients (5%) stage IIIB MM and 1 patient (3%) stage IIIA MM (Table 1). There were relatively more patients with pyrexia in stage III compared to stage IV, still the difference was not statistically significant (chi-square test: p = 0.077). The median and the mean pre-treatment values for LDH were slightly higher in patients without pyrexia (205 and 263 U/l) compared to patients who developed pyrexia (194 and 217 U/l). This difference was not statistically significant (t-test: p = 0.14). The study was not intended to evaluate the impact of pyrexia on the outcome of BRAF/MEK inhibitor combination therapy.

## Laboratory parameters

Using mixed model analysis, we developed a stepwise model for statistical analysis of the laboratory data. Model 1 was adjusted for time cluster, sex, and age. Model 2 (M2) was then adjusted only for time cluster and age and showed results that were statistically more significant. Model 3 (M3) contained the same covariates as model 2, but an additional term for the interaction between time clusters and the quartiles of the laboratory values was added to the equation. The results of the M3 analysis reached significance and showed also that the laboratory values have a time-dependent influence (Table 2). The interaction analysis of M3 then further showed the strength and significance of interaction between laboratory values and time cluster. These model analyses served to identify the parameters relevant for a pyrexia score, which were conspicuous in addition to the clinical experience.

When testing the quartiles of laboratory parameters (adjusted for time cluster and age) for significance, using mixed model analysis, the following results were obtained (Table 2). The analysis revealed a significant effect for elevated LDH (Model 1: F = 14.38; p< 0.0001; Model 2: F = 16.32; p < 0.0001; Model 3: F = 10.05; p = 0.002; Interaction M3: F = 4.96; P = 0.003). This effect was statistically significant for all 3 models, regardless of the time cluster. Furthermore, reduced leukocyte counts exerted an increasing influence as well (Model 1: F = 4.08; p = 0.045; Model 2: F = 5.74; p = 0.018; Model 3: F = 5.27; p = 0.023). For an elevated CRP, only the interaction between the quartiles of CRP and the fourth time cluster reached significance in the course of time (Interaction M3: F = 4.11; p = 0.008). A reduction of erythrocyte counts (Model 2: F = 4.88; p = 0.029) as well as reduced thrombocytes counts (Model 2: F = 11.96; p = 0.0007) demonstrated significance only in the model with adjustment for time cluster and age. Analysis of y-GT, AST, ALT, and S100 did not reveal statistically significant results.

**Table 1. Descriptive statistics of the first data set, patient´s tumor stage at time of therapy initiation.**

|  | IIIA | IIIB | IIIC | IV |
|---|---|---|---|---|
| **Pyrexia** | 0 | 1 | 5 | 8 |
| **no pyrexia** | 1 | 1 | 2 | 20 |
| *Total* | *1* | *2* | *7* | *28* |

**Table 2. Mixed model analysis of laboratory parameters: Model 1 with adjustment for time cluster, age and sex; Model 2 with adjustment for time cluster and age; Model 3 with adjustment for time cluster, age and interaction of laboratory value quartile with time cluster; Interaction term model M3 indicates strengths and significance of interaction between laboratory values and time cluster.**

| Value Quartiles | Model 1 | | Model 2 | | Model 3 | | Interaction Term Model for Model 3 | |
|---|---|---|---|---|---|---|---|---|
| | F | p | F | p | F | p | F | p |
| Leukocytes↓ | *4.08* | *0.045* | *5.74* | *0.018* | *5.27* | *0.023* | *0.15* | *0.927* |
| CRP↑ | 2.90 | 0.091 | 1.43 | 0.234 | 0.24 | 0.624 | *4.11* | *0.008* |
| LDH↑ | *14.38* | *< 0.001* | *16.32* | *< 0.001* | *10.05* | *0.002* | *4.96* | *0.003* |
| γ-GT↑ | 0.91 | 0.342 | 0.01 | 0.939 | 0.11 | 0.742 | 1.18 | 0.320 |
| AST↑ | 2.15 | 0.145 | 1.28 | 0.259 | 0.13 | 0.722 | 1.37 | 0.256 |
| ALT↑ | 3.49 | 0.064 | 1.34 | 0.249 | 1.18 | 0.281 | 2.21 | 0.090 |
| S 100↑ | 0.04 | 0.839 | 0.58 | 0.449 | 0.55 | 0.461 | 1.81 | 0.151 |
| Erythrocytes↓ | 2.99 | 0.086 | *4.88* | *0.029* | 3.85 | 0.052 | 0.91 | 0.439 |
| Thrombocytes↓ | *12.64* | *0.0005* | *11.96* | *0.0007* | *7.18* | *0.0085* | 2.33 | 0.061 |

– = model did not reach convergence; p = level of significance; F = test statistic of mixed models

↓ Decrease of absolute laboratory values

↑ Increase of absolute laboratory values

### Pyrexia scores determined by the algorithm approach 5.0

Based on the results obtained by our statistical analysis, lactate dehydrogenase (LDH) was added to the other three laboratory values: C-reactive protein CRP, leukocyte count and thrombocyte count. Fig 2 describes the general structure of the algorithm developed for the calculation of the pyrexia score.

Fig 3 demonstrates graphically the transformation of normalized patient's laboratory values into pyrexia scores. For this patient, laboratory values were determined at three clinical visits after start of therapy and till development of pyrexia. In the semi-logarithmic display, CRP and LDH values rise nearly linearly with time (Fig 3 and 3A). Likewise, in this patient leukocyte counts seem to fall linearly as well on a semi-logarithmic scale. After diagnosis of pyrexia and treatment interruption, values returned to normal range within 10 days. The pyrexia score

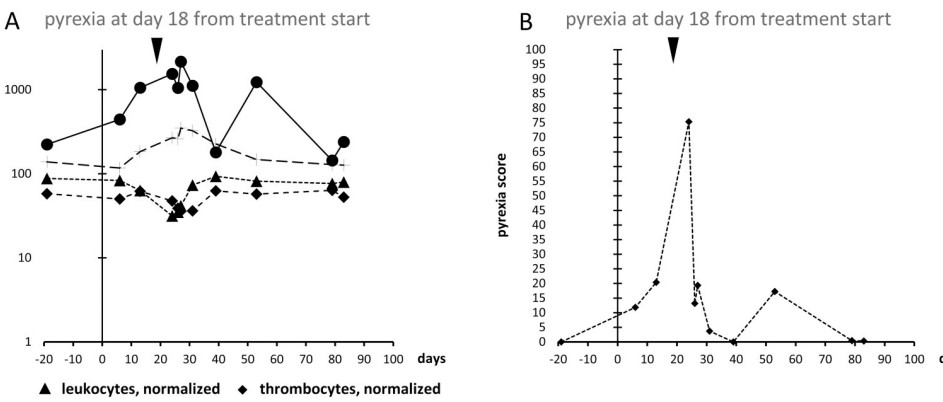

**Fig 3.** Transformation of individual and normalized laboratory values (A) of one patient with pyrexia into pyrexia scores (B) by algorithm 5.0. Inversed triangles indicate start of pyrexia. CRP was normalized to 100 = 2.5 mg/l (value CRP patient/2.5 * 100), LDH was normalized to 100 = 125 U/l, leukocyte count was normalized to 100 = 6.5 /nl (for male) & 6 /nl (for female), thrombocyte count was normalized to 100 = 275 /nl.

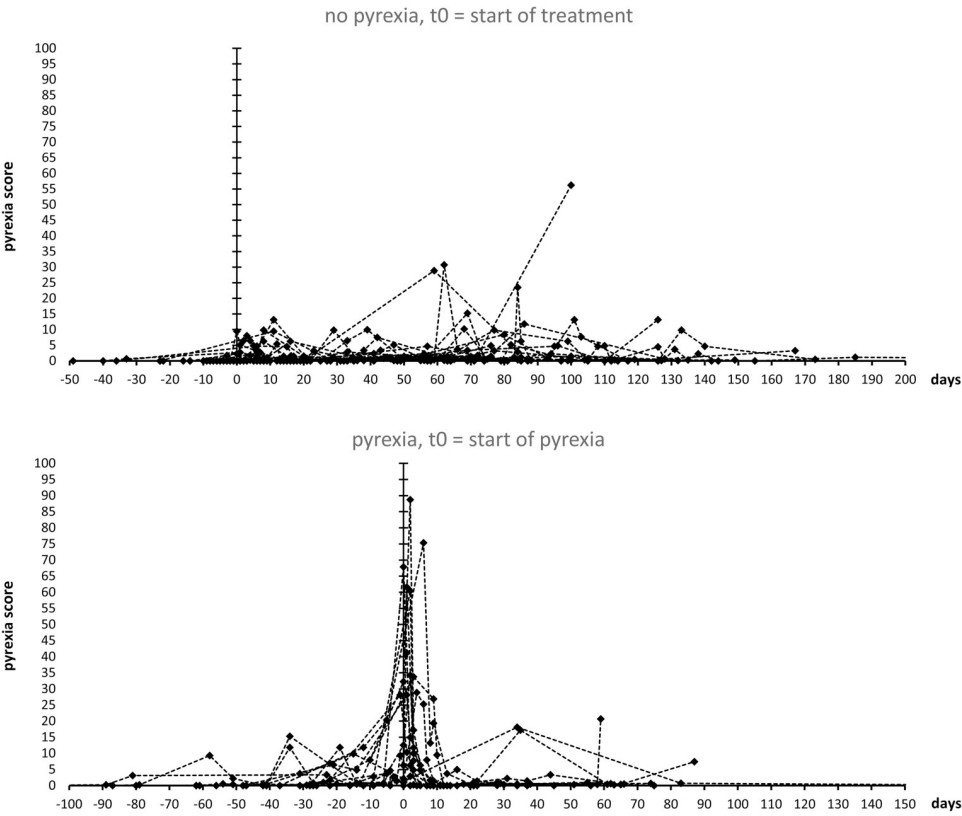

**Fig 4. Representative graphic displays of pyrexia scores by algorithm 5.0 of all analyzed patients over time; First data set, used for optimization of the pyrexia score algorithm.** All patients without pyrexia, day 0 indicates start of treatment. All patients with pyrexia, day 0 indicates start of pyrexia.

primarily assesses changes of analysed laboratory values only in the expected direction, i.e., rise of CRP and LDH and fall of thrombocytes and leukocytes. Absolute values are only used for defining thresholds (see materials and methods section). For the displayed patient, calculated pyrexia scores rise and obtain a maximum value after the onset of pyrexia (Fig 3 and 3B). Although CRP and LDH levels remain elevated and leukocyte and thrombocyte counts remain depressed shortly after the onset of pyrexia, the pyrexia score returns to low values immediately after the maximum as no further increase of CRP or LDH and no further decrease of leukocytes and thrombocytes occur in the patient after the onset of pyrexia and discontinuation of BRAF/MEK-inhibition.

Fig 4 shows a graphic display of pyrexia score time series using algorithm version 5.0 for all patients at all available data points. For patients without pyrexia, day 0 indicates the start of BRAF/MEK-inhibitor therapy, while for patients with pyrexia, day 0 indicates the start of pyrexia symptoms. The graphic display demonstrates that in most pyrexia patients, maximum pyrexia scores are obtained at or after the onset of pyrexia while most patients without pyrexia only show low score values at laboratory assessment dates. The observation that maximum PS score values in most pyrexia patients are found around the documented date of pyrexia and not at the exact date of pyrexia can be explained by both, the retrospective nature of the study and the calculation of the pyrexia score. The retrospective nature of the study implies that laboratory values most often were obtained after the onset of pyrexia while some laboratory assessments were obtained some days before the onset of pyrexia. Moreover, as the pyrexia score is

**Table 3. Values of variables, normalized for pyrexia score algorithm 5.0.**

| Values at t0 | | | | | | | |
|---|---|---|---|---|---|---|---|
| Leukocytes | | Thrombocytes | | LDH | | CRP | |
| F1 | T1 | F2 | T2 | F3 | T3 | F4 | T4 |
| 1.05 | 100 | 1.02 | 90 | 0.80 | 80 | 0.70 | 800 |
| FS1 | | FS2 | | FS3 | | FS4 | |
| 1.69 | | 3.24 | | 2.25 | | 2.25 | |
| **Values at t-1** | | | | | | | |
| Leukocytes | | Thrombocytes | | LDH | | CRP | |
| F5 | T5 | F6 | T6 | F7 | T7 | F8 | T8 |
| 1.10 | 120 | 1.10 | 90 | 0.80 | 80 | 0.70 | 500 |
| FS5 | | FS6 | | FS7 | | FS8 | |
| 1.44 | | 4.84 | | 2.25 | | 1.44 | |

| Factors for simultaneous change | | | | | | | CRP-gate at t0 |
|---|---|---|---|---|---|---|---|
| **Values at t0** | | | | | | | |
| Leu.-Thr.- | Leu.-LDH+ | Leu.-CRP+ | Thr.-LDH+ | Thr.-CRP+ | LDH+CRP+ | | CT 300 |
| OA1 | OA2 | OA3 | OA4 | OA5 | OA6 | | GC 0.1 |
| 2.0 | 1.4 | 1.2 | 1.4 | 1.0 | 1.1 | | |

| Infection signature at t = 0 | | | | | | | |
|---|---|---|---|---|---|---|---|
| Leukocytes | | CRP | | Thrombocytes | | LDH | |
| IC1 | IM1 | IC2 | IM2 | IC3 | IM3 | IC4 | IM4 |
| 0.90 | 100 | 0.90 | 200 | 0.90 | 100 | 0.90 | 120 |

based on changes of laboratory values in the expected direction, scores determined during pyrexia symptoms but after interruption of treatment may be lower than scores obtained shortly before onset of symptoms.

As described by material and methods, the pyrexia score algorithm was optimized through an iterative approach by calculation of ratios MWP/(M0P+MWP) and MWP14/(M0P + MWP) and by visualization of all pyrexia scores as depicted by Fig 4. The algorithm is based on heuristic assumptions and the optimization process can only be considered approximative. The resulting normalized values of variables for algorithm 5.0 are shown by Table 3. It is of note that variables FS which denote a weighting parameter for laboratory values at t0 and t-1 display highest values for the thrombocyte count, followed by values for LDH. Both laboratory values have also demonstrated highest F-values and highest levels of significance in the mixed model analysis using time clusters. Variables OA for simultaneous change demonstrated the highest value for simultaneous reduction of thrombocyte and leukocyte counts. The addition of additive factors OA in versions 4.0 and 5.0 adds a stronger spread between patients with and without pyrexia in the graphic display and addition of a CRP threshold and introduction of an infection signature into the algorithm (see materials and methods section) leads to a smoothing of pyrexia scores not associated with pyrexia in the graphic display. The intention of the pyrexia score algorithm was not only to provide an additional approach for statistical validation of the putative laboratory signature but also to provide a tool for visualization of laboratory changes associated with pyrexia for the treating clinician as a future clinical decision support system. In this regard, version 5.0 demonstrated best graphical visualization and was further analysed with binary logistic regression. The processing of the pyrexia score as well as the anonymized datasets containing time series of leukocyte and thrombocyte counts as well

**Table 4. ANOVA-analysis of pyrexia-score 5.0.**

| Pyrexia score 5.0 | Values | Mean | Median | Min | Max | Std.error | Std.dev. | IQR |
|---|---|---|---|---|---|---|---|---|
| *Non-pyrexia patients* | 255 | 2.151 | 0.270 | 0.000 | 56.246 | 0.331 | 5.285 | 1.690 |
| *Pyrexia patients* | 162 | 6.604 | 0.651 | 0.000 | 88.741 | 1.110 | 14.113 | 6.280 |

as of CRP and LDH serum levels of all patients have been deposited as open access in the data-dryad.org data repository as an excel file (https://doi.org/10.5061/dryad.xpnvx0kjj).

## Association of pyrexia-score and pyrexia

ANOVA analysis showed that there was a significant difference between the two groups (pyrexia/no pyrexia) regarding the pyrexia-score (Table 4). The pyrexia-score differentiated well between patients with and without pyrexia (F = 20,8; p = <0,0001) and was significantly higher in patients with pyrexia. In this regard, the pyrexia score outpaced single laboratory values. The pyrexia-score displays a better variance clarification than the single-value analysis and has, therefore, a significantly better predictive power (in the sense of diagnosing pyrexia) than individual values, due to the specific weighting of the individual elements through the algorithm.

Pyrexia scores of pyrexia patients were divided into quartiles and the odds ratios were determined using logistic regression in order to define a putative cut-off value for pyrexia. The 4[th] quartile of PS5.0 of patients with pyrexia demonstrated an odds-ratio of 3.446 (95% CI: 1.907–6.227) for development of pyrexia at a value > = 6.280 (Tables 4 and 5). Binary logistic regression analysis (generalized, mixed model) showed a significant predictive value of the pyrexia-score on the diagnosis of pyrexia (fixed effects pyrexia-score: F = 6.24; p = 0.013). ROC analysis showed a good relationship between sensitivity and specificity of the pyrexia-score (AUC ROC-curve = 0.9480). Through threshold analysis for each score of each patient, it could be predicted when there was a significant increase in the risk of diagnosing pyrexia and by how much the risk increased compared to patients with a lower score than the respective threshold.

## Validation of the pyrexia score algorithm with an independent data set

We tested the predictive (diagnostic) power of the pyrexia score on an additional data set from the University Hospital of Cologne and from the University Hospital of the RWTH Aachen (Fig 5). Of the 28 cases, the pyrexia score algorithm 5.0 correctly diagnosed the development of pyrexia or no pyrexia in 23 cases using the threshold value of > = 6.280. In 5 cases, the score led to a prediction that did not match the clinical course. In the sensitivity/specificity analysis (ROC), the score reached an area under curve (AUC) of 0.92. In 16 cases, the absence of pyrexia and in 7 cases, the occurrence of pyrexia were correctly diagnosed. 1 case of pyrexia

**Table 5. Odds-ratio of Pyrexia-score 5.0; P0 = Patients without pyrexia, Quartile = Quartile of pyrexia score of patients with pyrexia.**

| Score | Odds-ratio | 95% Wald CI | Pyrexia Score |
|---|---|---|---|
| *Quartile 1 vs P0* | 1.423 | 0.778–2.604 | 0.014 < 0.468 |
| *Quartile 2 vs P0* | 1.324 | 0.709–2.469 | 0.468 < 1.440 |
| *Quartile 3 vs P0* | 1.220 | 0.669–2.224 | 1.440 < 6.280 |
| *Quartile 4 vs P0* | 3.446 | 1.907–6.227 | > = 6.280 |

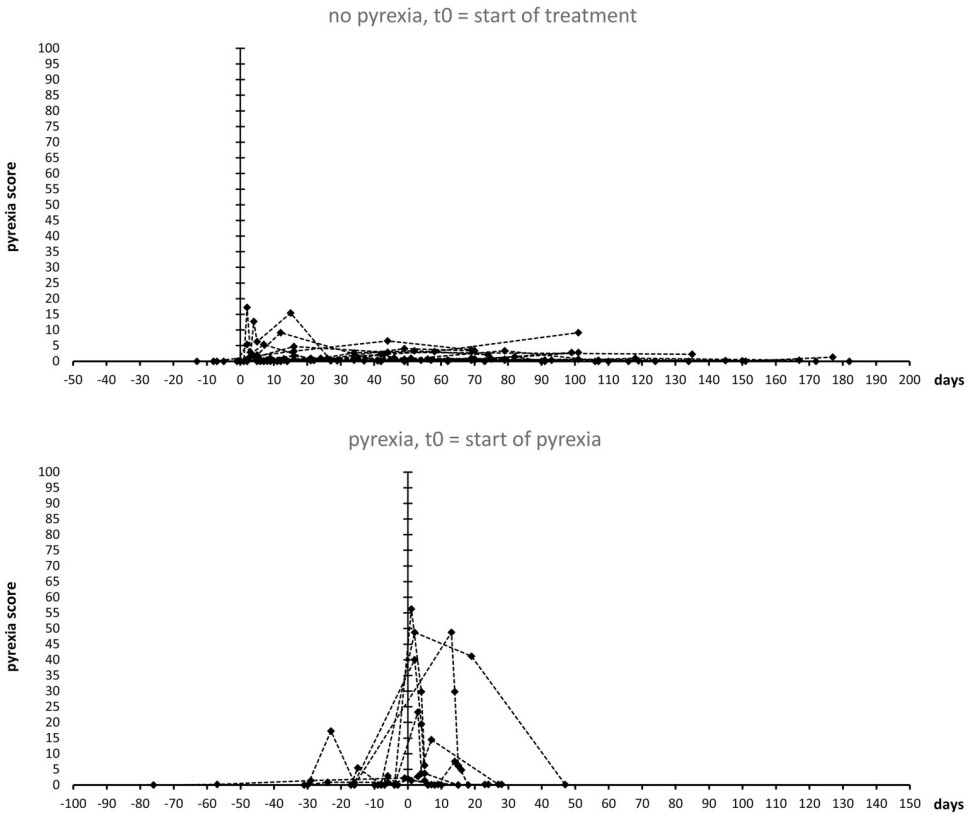

**Fig 5. Representative graphic displays of pyrexia scores by algorithm 5.0 of all analyzed patients over time; second independent data set.** All patients without pyrexia, day 0 indicates start of treatment. All patients with pyrexia, day 0 indicates start of pyrexia.

could not be diagnosed by the score and in 4 cases the score indicated the development of pyrexia, which then did not occur.

## Discussion

Pyrexia is by far the most common of all adverse events in patients with malignant melanoma who receive combination treatment with the BRAF-inhibitor dabrafenib and the MEK-inhibitor trametinib [5, 11]. Pyrexia may also occur during treatment with BRAF/MEK combinations vemurafenib/cobimetinib as well as encorafenib/binimetinib, albeit at a considerably lower frequency. Median time of onset is 19 days (range 1–82 days), and median duration is 9 days [11]. In multiple studies, combination therapy with dabrafenib and trametinib has shown improved survival rates and progression-free survival for patients afflicted with BRAF-V600--mutant metastatic malignant melanoma [6, 7, 11] as well as for other patients with non-melanoma cancers [22, 23].

Although life-threatening toxicities associated with BRAFi and MEKi toxicities are rare and uninterrupted treatment is advised in case of mild toxicities, in case of moderate to severe side effects, treatment interruption is advised. Till date, early management of pyrexia has been performed by patient education for the prodromes as well as by interruption or dose reduction of BRAF/MEK inhibitors at the very first symptoms [11]. Recent studies demonstrated that the incidence of pyrexia is particularly high in the early stages of therapy and that temporary interruption of dabrafenib or trametinib is the most effective way to manage pyrexia [24–26].

Adapted pyrexia management seems to reduce the severity of pyrexia and to enhance treatment adherence [26]. Early and precise confirmation of pyrexia and differentiation of pyrexia from other febrile conditions is, therefore, mandatory.

By visual analysis of routine laboratory values of melanoma patients developing pyrexia under BRAF/MEK inhibition, we observed that CRP rises and leukocyte as well as thrombocyte counts drop simultaneously before and during pyrexia. Our retrospective statistical analysis of a cohort with and without pyrexia under dabrafenib plus trametinib therapy using time clusters further revealed that elevated LDH levels under treatment were also associated with pyrexia. In order to prove whether a putative four laboratory value signature of pyrexia, i.e., the simultaneous rise of CRP and LDH as well as drop of leukocyte and thrombocyte counts, correlated with the development of pyrexia, we designed an algorithmic approach that calculates a pyrexia score based on the predicted laboratory value changes. The algorithm itself was designed by a heuristic approach combined with an iterative optimization procedure [20].

Statistical analysis demonstrated that the individual parameters by themselves displayed only modest significance and had low predictive power compared to the pyrexia scores calculated by the proposed algorithm. This underlines the importance of the interaction between the laboratory values as well as of the time-dependent changes. The score's predictive performance with regard to the diagnosis of pyrexia derives from the specific weighting of the individual elements through the algorithm and its course over time. Maximum pyrexia score values in patients correlated with pyrexia and statistical analysis demonstrated that the score outplays each single laboratory value with regards to its diagnostic performance. Although elevated pyrexia scores could be measured in few patients already before the onset of pyrexia, highest score values were found mainly after the onset of pyrexia. Therefore, the basic use of the pyrexia score seems to be the confirmation of drug-induced pyrexia under dabrafenib plus trametinib therapy. True prediction of pyrexia in the clinical sense of being able to foresee the development of pyrexia by the score seems to be impaired, on the one hand, by missing laboratory values determined only few days before the onset of pyrexia. On the other hand, graphical displays of laboratory time series suggest that leukocyte and thrombocyte counts might fall only shortly before the onset of fever (see Figs 1 and 3), thus limiting the diagnostic window during which the pyrexia score might clinically predict the onset of pyrexia.

Until now the exact cause of pyrexia development is still unclear but the release of proinflammatory proteins and cytokines seem to play a central role as it has been noted before [11–13, 15–17]. In patients under therapy with BRAF/MEK inhibitors but also by in-vitro experiments it could be demonstrated that the BRAF inhibitor dabrafenib induces IL-1beta (IL-1β) to a greater extent than the other BRAF inhibitors which are associated less frequently with pyrexia. Moreover, it could be demonstrated that the degree of IL-1beta release under dabrafenib displays individual variability which might explain why only a subset of patients develop pyrexia [15, 17]. Other side effects of BRAF inhibition include conditions with hyperproliferation of keratinocytes leading to the development of a spectrum of hyperkeratotic conditions including actinic keratosis, Grover's disease, cutaneous squamous cell carcinoma and acneiform eruptions. A peak in these conditions is seen between weeks 8 and 36 of treatment [27]. A paradoxical activation of the MAPK pathway by BRAF inhibitors was proposed as a causative factor. It was hypothesized that in BRAF wild-type myeloid cells, paradoxical MAPK signaling activation might be induced by BRAF-inhibitors and that this mechanism might cause pyrexic events [28]. Although it has been shown that the BRAF-inhibitor dabrafenib induces inflammasome activation and IL-1beta release [17], the role of MAPK signaling in inflammasome activation is not clear. The addition of a MEK-inhibitor to BRAF inhibition does not seem to reduce the frequency of pyrexia which might suggest additional molecular mechanisms linking BRAF inhibition to immune cell activation. It has been demonstrated that

KRAS overexpression may induce inflammasome activation in myeloid cells by the KRAS/RAC1/ROS/NLRP3/IL-1β-axis [29] which might suggest a MEK-independent pathway.

Although the observed laboratory signature of pyrexia is not able to identify the cause of treatment-induced pyrexia by itself, the presented data and the analysis by the pyrexia score algorithm suggest that thrombocytes might play an additional role during pyrexia and that the induction of pyrexia might follow a self-amplifying process leading to a strong increase of proinflammatory factors. It could be shown that thrombocytes are able to boost IL-1beta production in inflammasome-activated monocytes and that platelets are an additional source of IL-1beta [30]. The simultaneous drop of leukocytes and thrombocytes might be induced by aggregation of thrombocytes and granulocytes followed by sequestration in the liver and spleen as described in virus-induced hemorrhagic fever [31, 32]. Interestingly in hemorrhagic fever, LDH serum levels seem to correlate with severity of disease [33].

Main limitations of the presented study are the small number of patients involved and the inhomogeneous times of laboratory data assessments, which have to be attributed to the retrospective nature of the study. In many pyrexic patients, CRP values were only determined after pyrexia had occurred, so we often lacked data points for the pre-pyrexic time cluster. Our study did not identify a laboratory value which could clinically predict pyrexia before treatment start as no predictive laboratory value or combination of values were identified for the 1st time cluster. In order to address this important issue which would greatly impact on treatment decisions in melanoma patients, it might be necessary to focus more on cytokines and on polymorphisms of cytokines associated with the inflammasome function.

Another limitation of the presented study which also stems from its retrospective nature lies in the diagnosis of pyrexia which was only based on the physician's clinical documentation but did not follow a structured diagnostic algorithm that would have been able to exclude other causes of fever besides pyrexia and infection. Tumor fever, sarcoidosis-like reactions as well as pancreatitis may occur under dabrafenib plus trametinib treatment and might have induce fever in the analysed patients.

The structure of the data only allows a prediction regarding the diagnosis of pyrexia but does not allow a more precise temporal differentiation of the exact time of onset of pyrexia in individual patients. We recommend performing a prospective study with a larger number of patients using fixed time clusters that can be set based on our preliminary results. The analysis of inflammatory parameters should also be expanded including IL-1β. Furthermore, as the biomarker LDH correlates with the tumor burden in melanoma patients, it might be necessary quantifying tumor burden independently by quantitative liquid biopsy based on the BRAF mutation. This would help identifying response to the treatment as a confounding factor on LDH levels.

It must be taken into account that the pyrexia score was optimized in order to discriminate between patients with and without pyrexia and to identify the onset of pyrexia. As a consequence, high score values indicate development of pyrexia but do not necessarily provide a measurement of the severity of pyrexia. When leukocyte and thrombocyte numbers do not diminish further or when CRP and LDH do not increase any more, the pyrexia score quickly falls although the associated symptoms of pyrexia, such as fever, leukopenia or thrombocytopenia, might still be severe. Therefore, the pyrexia score may not be regarded as a decision tool for corticosteroid treatment of pyrexia. In this respect, fever CTCAE grading as well as absolute leukocyte and thrombocyte counts together with their dynamics might be more informative.

Still, by opting for an algorithmic approach based on heuristic assumptions [18] we were able to demonstrate the statistical significance of the observed and postulated four-value laboratory signature of pyrexia. From a methodological standpoint, defining algorithms based on

heuristic assumptions may be very useful to assess clinical hypotheses with real world patient data which, unfortunately, are often incomplete and inhomogeneous. Moreover, laboratory values of patients in the routine practice will be influenced by many factors that treating physicians normally take into account when assessing the patient's laboratory results, but which might reduce the validity of any traditional statistical evaluation. An algorithm may integrate information from different sources, thus offering a much-needed standardized metric. To the best of our knowledge, this work is the first study proposing a pyrexia score for diagnosis of pyrexia. The proposed algorithm, possibly in a more elaborate version, may help clinicians monitor BRAF/MEK-inhibitor treatment more efficiently and thus optimize treatment outcome and increase patients' quality of life.

## Author Contributions

**Conceptualization:** Albert Rübben, Marike Leijs.

**Data curation:** Hannah Schaefer.

**Formal analysis:** Hannah Schaefer, Albert Rübben, André Esser, Marike Leijs.

**Investigation:** Hannah Schaefer, Oana-Diana Persa, Marike Leijs.

**Methodology:** Hannah Schaefer, Albert Rübben, André Esser, Marike Leijs.

**Software:** Albert Rübben, André Esser, Arturo Araujo.

**Supervision:** Albert Rübben, Marike Leijs.

**Validation:** Hannah Schaefer, Albert Rübben.

**Visualization:** Hannah Schaefer, Albert Rübben, André Esser.

**Writing – original draft:** Hannah Schaefer.

**Writing – review & editing:** Albert Rübben, André Esser, Arturo Araujo, Oana-Diana Persa, Marike Leijs.

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
