## [Decision Letter · Decision Letter 0]

21 Jun 2022

PONE-D-22-11421A distinct four-value blood signature of pyrexia under combination therapy of malignant melanoma with BRAF/MEK-inhibitors evidenced by an algorithm defined pyrexia scorePLOS ONE

Dear Dr. Rübben,

Thank you for submitting your manuscript to PLOS ONE. After careful consideration, we feel that it has merit but does not fully meet PLOS ONE’s publication criteria as it currently stands. Therefore, we invite you to submit a revised version of the manuscript that addresses the points raised during the review process.

Please adapt your manuscript according to the suggestions of the reviewers. Please discuss the reasons where this might not be possible.

We look forward to receiving your revised manuscript.

Kind regards,

Michael C Burger, M.D.

Academic Editor

PLOS ONE

Journal Requirements:

2.Please review your reference list to ensure that it is complete and correct. If you have cited papers that have been retracted, please include the rationale for doing so in the manuscript text, or remove these references and replace them with relevant current references. Any changes to the reference list should be mentioned in the rebuttal letter that accompanies your revised manuscript. If you need to cite a retracted article, indicate the article’s retracted status in the References list and also include a citation and full reference for the retraction notice.

"The authors have declared that no competing interests exist. . A.R. has received travel grants and consulting remunerations from Roche, MSD, BMS, Amgen and Novartis as well as a research grant from Novartis unrelated to the presented study."

Reviewers' comments:

Reviewer's Responses to Questions

**Comments to the Author**

1. Is the manuscript technically sound, and do the data support the conclusions?

Reviewer #1: Yes

Reviewer #2: Partly

2. Has the statistical analysis been performed appropriately and rigorously? 

Reviewer #1: I Don't Know

Reviewer #2: I Don't Know

3. Have the authors made all data underlying the findings in their manuscript fully available?

Reviewer #1: Yes

Reviewer #2: Yes

4. Is the manuscript presented in an intelligible fashion and written in standard English?

Reviewer #1: No

Reviewer #2: Yes

5. Review Comments to the Author

Reviewer #1: Schaefer et al. submitted an original manuscript dealing with a biomarker-model to Plos ONE. This model can predict pyrexia in BRAF/MEK inhibitor-treated melanoma patients when analysing on-treatment changes in stadard lab values. The content and conclusions are sound and interesting, but the manuscript needs to be improved in regards of layout and data presentation. Figures legends appear at random places, and tables do not hat a footnote explaining abbrevation. Design of Table 1 is horrible. Usage, language and style are ok but could be better.

Reviewer #2: The authors analyzed laboratory changes to predict combi-DT-induced pyrexia and showed that a pyrexia score calculated from CRP, LDH, leukocyte and thrombocyte numbers correlated with the appearance of pyrexia. Although this kind of study may potentially be interesting, the statistical analysis in this paper is difficult to follow and I am not confident that I am evaluating this paper correctly. I have a fear that most readers will be not able to fully understand the content as well as I did not. Considering that this retrospective study involves a very small number of patients, although the authors made their own settings with respect to the time clusters established in this study, it is possible that this just happened to make a significant difference in the analysis of the present data set.

My suggestions are as follows;

#1. Of the 28-patient data set used to validate the pyrexia score algorithm, patients who received treatment other than combi-DT should be excluded.

#2. Although the authors described that “most patients without pyrexia only show low score values at laboratory assessment dates“, can the author provide the cut-off value for pyrexia score?

#3. As shown in the limitation, figures 4 and 5 show that the PS reaches its maximum value in the blood collection immediately “after” the pyrexia events, while the PS is low before the pyrexia events in the most cases. Therefore, it is difficult to say that the score is predictive of fever.

#4. Is there a correlation between the severity of pyrexia and PS? Could the severity of pyrexia be classified according to some criterion such as CTCAE? If the pyrexia is more severe in patients with high PS, it may allow for the early administration of corticosteroids to such patients.

#5. I guess that in cases with high baseline LDH levels, whose tumor volume is assumed to be high, it may be difficult to differentiate combi-DT-induced pyrexia from tumor fever. Were the authors able to exclude tumor fever from the differential diagnosis? Was there any difference in baseline LDH or Stage at the treatment initiation between the pyrexia and non-pyrexia groups?

#6. As described in the manuscript, more than half of the patients used corticosteroids as treatment for pyrexia. Since corticosteroids have a significant impact on white blood cell counts, I think it would not be appropriate to analyze patients who did and did not receive this treatment together.

6. PLOS authors have the option to publish the peer review history of their article (what does this mean?). If published, this will include your full peer review and any attached files.

Reviewer #1: No

Reviewer #2: No

---

## [Author Response · Author response to Decision Letter 0]

4 Aug 2022

Response to Reviewers’ comments and suggestions:

Manuscript: PONE-D-22-11421

Reviewer 1: 

Schaefer et al. submitted an original manuscript dealing with a biomarker-model to Plos ONE. This model can predict pyrexia in BRAF/MEK inhibitor-treated melanoma patients when analysing on-treatment changes in standard lab values. The content and conclusions are sound and interesting, but the manuscript needs to be improved in regards of layout and data presentation. Figures legends appear at random places, and tables do not hat a footnote explaining abbreviation. Design of Table 1 is horrible. Usage, language and style are ok but could be better.

Answer: Thank you very much for this helpful comment. We agree that our manuscript needs improvement in terms of layout and data presentation and, therefore, we redesigned the figures with a uniform layout and simplified the tables as well as the data presentation within the text, referring especially to table 1.

Reviewer 2

1. The authors analyzed laboratory changes to predict combi-DT-induced pyrexia and showed that a pyrexia score calculated from CRP, LDH, leukocyte and thrombocyte numbers correlated with the appearance of pyrexia. Although this kind of study may potentially be interesting, the statistical analysis in this paper is difficult to follow and I am not confident that I am evaluating this paper correctly. I have a fear that most readers will be not able to fully understand the content as well as I did not. 

• Answer: Thank you for your valuable comment. We tried to clarify ambiguous formulations within the statistical description and discussion, especially with regard to time clusters and the predictive value (see also answers to comments no. 2 and 5).

The design of the algorithm has been described in detail in order to facilitate similar approaches by other researchers. In addition, in the revised manuscript we will provide not only the normalized laboratory values of all patients with respect to the pyrexia signature, but we will also include the calculations of the algorithm within an excel spreadsheet. This approach should enable the interested reader to reproduce the statistical analysis and to better understand the algorithmic approach.

2. Considering that this retrospective study involves a very small number of patients, although the authors made their own settings with respect to the time clusters established in this study, it is possible that this just happened to make a significant difference in the analysis of the present data set.

• Thank you very much for this comment. 

We are well aware of the relatively low number of patients and, hence, recommended a prospective study with more participants. 

The enclosed post hoc power analysis indicates that to obtain a significantly stronger statistical impact, we would need approx. 100 patients (see graph in the uploaded file).

Nevertheless, the evaluation of the algorithm with an additional independent data set of 28 patients confirmed the usefulness and the statistical power of the algorithmic approach. The total sample size including the second data set is 64 patients. 

With respect to the time cluster analysis, we would like to place emphasis on the step-wise evolution of the study: As described in the manuscript, the initial observation in single patients suggested a concurrent decrease in leukocyte and thrombocyte counts together with an elevation of serum C-reactive protein during pyrexia.

In order to verify or falsify the hypothesis that this might represent a laboratory signature of pyrexia, our PhD-student was instructed to analyse patient’s laboratory data together with our statistician from the Department of Occupational, Social and Environmental Medicine in the classical way. They conceived the time cluster approach in order to control for the temporal dependence of pyrexia. As stated in the manuscript, this time cluster approach was also used for patients without pyrexia based on the determination of the mean time value of the occurrence of pyrexia in our data set, thereby assuming a time at which a pyrexia might have occurred in these patients. We have performed this approach as it could be that the suspected laboratory values might change by a similar mode in time regardless of pyrexia.

Out of this analysis, unexpectedly, elevation of serum lactate dehydrogenase appeared as the strongest laboratory value associated with pyrexia. LDH was included in the quest for a pyrexia signature as time series in patients with pyrexia strongly suggested that during pyrexia LDH rises concurrently with CRP. The finding of an unexpected laboratory value associated with pyrexia, which subsequently also proved useful for pyrexia score analysis, demonstrates that time clusters were not chosen and adapted in order to gain significance within the statistical analysis. The design of the pyrexia score algorithm was preconceived only later as we still assumed that the performed statistical analysis did not prove that the synchrony of the laboratory changes by itself was a characteristic of pyrexia.

We understand that the way by which we analysed the available laboratory data, first by statistics with time clusters and then with an algorithm leading to a score, might appear complicated and non-orthodox. On the other hand, it reflects the gain in comprehension derived from a stepwise approach.

3. Of the 28-patient data set used to validate the pyrexia score algorithm, patients who received treatment other than combi-DT should be excluded. 

• Answer: Thank you very much for your feedback. We totally agree that it is preferable to omit patients treated with other BRAF/MEK inhibitors as pyrexia occurs more frequently with dabrafenib treatment compared to other BRAF inhibitors and as the first data set included only patients treated with dabrafenib plus trametinib. 

The second data set used for validation contained four patients treated with encorafenib plus binimetinib. For the revised manuscript we have omitted these four patients and we have recalculated the statistics with a dataset without these four patients and with a data set where these four patients were replaced by four patients which were treated with dabrafenib plus trametinib. In order to avoid any bias, three patients without pyrexia were selected chronologically from our patient treatment list and the next first patient with pyrexia was included in order to substitute for the one pyrexia patient treated with encorafenib plus binimetinib. 

In the sensitivity/specificity analysis (ROC) with the original second data set which contained the four patients treated with encorafenib plus binimetinib, the pyrexia score algorithm reached an area under curve (AUC) of 0.83. Recalculation of the area under the curve with the second data set without the four patients demonstrated an area under curve of 0.85 (thus a better fit) and recalculation with the second data set without the four patients treated with encorafenib plus binimetinib but with four additional patients treated with dabrafenib plus trametinib even revealed an area under curve of 0.92. Although statistics already improved by omitting patients treated with encorafenib plus binimetinib, we have submitted the revised manuscript with the new dataset with four new patients in order to maintain the patient numbers. Moreover, together with the revised manuscript we have deposed all anonymized and normalized patient laboratory value time series for leukocyte counts, thrombocyte counts, CRP and LDH. Thereby we not only allow independent verification of our algorithm but also give the possibility to modify or optimize the algorithm or to analyse the data by other statistical procedures. Exchanging the four patients treated with encorafenib plus binimetinib by four new patients treated with dabrafenib plus trametinib provides a larger dataset for independent verification.

Accordingly, we also modified the title to “A distinct four-value blood signature of pyrexia under combination therapy of malignant melanoma with dabrafenib and trametinib evidenced by an algorithm-defined pyrexia score‘‘ replacing “BRAF/MEK inhibitors” with “dabrafenib and trametinib” as the four-value blood signature might be only correlated to dabrafenib/trametinib-induced pyrexia.

4. Although the authors described that “most patients without pyrexia only show low score values at laboratory assessment dates“, can the author provide the cut-off value for pyrexia score?

• Answer: A cut-off value for the pyrexia score is reported by tables 4 and 5. We have clarified this point in the revised manuscript (lines 481-484).

5. As shown in the limitation, figures 4 and 5 show that the PS reaches its maximum value in the blood collection immediately “after” the pyrexia events, while the PS is low before the pyrexia events in the most cases. Therefore, it is difficult to say that the score is predictive of fever.

• Answer: Thank you very much for this important comment. Predictive in a statistical sense means the probability that a person has a condition given a positive test result. Therefore, the pyrexia score is predictive as it is able to correctly diagnose pyrexia. A predictive test in the clinical sense allows identifying patients with an enhanced risk for development of a condition. In our study we did not detect laboratory values obtained for the time cluster 1 and 2 which might have demonstrated a correlation with subsequent development of pyrexia. In future studies it might be helpful to focus more on cytokines and on polymorphisms of cytokines associated with the inflammasome function in order to identify clinically predictive factors. In the revised manuscript we have clarified that “predictive” refers to the power of diagnosing pyrexia and that true clinical prediction of pyrexia before pyrexia occurs is not possible with the score (see lines 477-490 and 553-562).

6. Is there a correlation between the severity of pyrexia and PS? 

7. Could the severity of pyrexia be classified according to some criterion such as CTCAE? 

• Answer: Thank you very much for these two important comments. 

As stated, pyrexia in our retrospective study was defined as an oral temperature of 38.5°C (≥101.3°F) or higher, in the absence of any clinical or microbiological evidence of infection. Due to the retrospective nature of the study, we could not classify the severity of pyrexia as we had to rely on the clinical documentation which was often based on patients’ personal accounts which did not allow for a reliable distinction between different CTCAE grades. 

The pyrexia score itself does not evaluate the severity or the duration of fever but only analyses laboratory changes. Still, on the basis of the putative molecular mechanisms, as hypothesized in lines 556-586, we would expect a positive correlation between the severity of pyrexia and the maximal score values. This could be assessed in a future prospective study.

8. If the pyrexia is more severe in patients with high PS, it may allow for the early administration of corticosteroids to such patients.

Answer: As to the previous comments, we could not reliably grade pyrexia due to the retrospective nature of the study. Moreover, it must be taken into account that the score was optimized in order to discriminate between patients with and without pyrexia and to identify the onset of pyrexia. As a consequence, high score values indicate development of pyrexia but do not necessarily reflect severity of pyrexia. Moreover, when leukocyte and thrombocyte numbers do not diminish further or when CRP and LDH do not increase any more, the pyrexia score quickly falls although the associated symptoms of pyrexia, such as fever, leukopenia or thrombocytopenia, might still be severe. 

Nevertheless, the question itself, when to administer corticosteroids to pyrexia patients, is of great clinical significance. Our score should help to better identify pyrexia in the first place. Once pyrexia is very likely, we would suggest relying on absolute leukocyte and thrombocyte counts as well as on their dynamics for deciding whether to initiate corticosteroid treatment. This is what we currently do in our clinical routine, but we may not provide any specific cut-off values as we have no reliable data for such a claim. We have included a discussion on this point in the revised manuscript (see lines 618-626).

9. I guess that in cases with high baseline LDH levels, whose tumor volume is assumed to be high, it may be difficult to differentiate combi-DT-induced pyrexia from tumor fever. Were the authors able to exclude tumor fever from the differential diagnosis?

• Answer: We totally agree that tumor fever might have led to the erroneous diagnosis of treatment-induced pyrexia. We have discussed this possibility in our revised manuscript (see lines 603-608). In addition, fever and elevated LDH or CRP under treatment might also be associated with a sarcoidosis-like reaction as well as with pancreatitis, both representing known side effects of BRAF/MEK inhibitor treatment. In our experience with malignant melanoma, both conditions seem to occur even more frequently as compared to tumor fever. As our study was retrospective, we have scored patients as having pyrexia only by the clinical assessment of the treating physician as documented in the clinical documentation or by the discharge summary. We have not reassessed each case for possible competing causes of fever as we did not want to introduce an additional bias by only selecting patients which fulfilled our personal speculations. We strongly believe that in a future prospective study where competing sources of fever in treated patients would be ruled out through a standardized approach, the association of the described laboratory signature with pyrexia should be even stronger. 

10. Was there any difference in baseline LDH or Stage at the treatment initiation between the pyrexia and non-pyrexia groups?

• Answer: Thank you very much for your question and pointing this out. In effect, the median and the mean normalized pre-treatment values for LDH were slightly higher in patients without pyrexia (163.9 and 210.6) compared to patients who developed pyrexia (155.0 and 173.9). This difference was not statistically significant (T-test: p=0.14). There were relatively more patients with pyrexia in stage III compared to stage IV, still the difference was not statistically significant (Chi2-test: p=0.077). We have included this information in the revised manuscript (see lines 354-362).

11. As described in the manuscript, more than half of the patients used corticosteroids as treatment for pyrexia. Since corticosteroids have a significant impact on white blood cell counts, I think it would not be appropriate to analyze patients who did and did not receive this treatment together.

• Answer: Thank you for your valuable input. Yes, severe pyrexia events led to treatment with glucocorticoids in 8 cases, but steroid treatment was only conducted after the clinical development of pyrexia and not before pyrexia. Therefore, bloodwork parameters, especially regarding white blood cell counts, were only modified by steroids after the beginning of pyrexia and thus were not included in the statistical analysis. Time clusters 1-4 only contain values before administration of glucocorticoids. Time cluster 4 contains the first laboratory value after pyrexia occurred for each patient which was obtained when the patient presented in the clinic with symptoms of pyrexia. At this time the patient has not yet received corticosteroids. In figures 4 and 5, highest pyrexia score values are often found after the onset of pyrexia, which is explained by the fact that all patients developed the symptoms at home and only rarely presented at the clinic on the same day of first symptoms. In the revised manuscript we have clarified this point (line 190).

In addition to the changes suggested by the reviewers we have deposited the processing of the pyrexia score as well as the anonymized datasets containing time series of leukocyte and thrombocyte counts as well as of CRP and LDH serum levels of all patients in the open access datadryad.org data repository as an excel file (https://doi.org/10.5061/dryad.xpnvx0kjj). 

We have further omitted statements with “data not shown”.

The authors declare no conflict of interest.

---

## [Editor Report · Decision Letter 1]

10 Aug 2022

A distinct four-value blood signature of pyrexia under combination therapy of malignant melanoma with dabrafenib and trametinib evidenced by an algorithm defined pyrexia score

PONE-D-22-11421R1

Dear Dr. Rübben,

We’re pleased to inform you that your manuscript has been judged scientifically suitable for publication and will be formally accepted for publication once it meets all outstanding technical requirements.

Kind regards,

Michael C Burger, M.D.

Academic Editor

PLOS ONE
---

## [Editor Report · Acceptance letter]

15 Aug 2022

PONE-D-22-11421R1 

A distinct four-value blood signature of pyrexia under combination therapy of malignant melanoma with dabrafenib and trametinib evidenced by an algorithm-defined pyrexia score 

Dear Dr. Rübben:

I'm pleased to inform you that your manuscript has been deemed suitable for publication in PLOS ONE. Congratulations! Your manuscript is now with our production department. 

Kind regards, 

on behalf of

Dr. Michael C Burger 

Academic Editor

PLOS ONE